# Phosphorus-Rich Ash from Poultry Manure Combustion in a Fluidized Bed Reactor

**Zdzisław Adamczyk [1], Magdalena Cempa [2],\* and Barbara Białecka [2]**

[1] Faculty of Mining, Safety Engineering and Industrial Automation, Department of Applied Geology, Silesian University of Technology, 44-100 Gliwice, Poland; zdzislaw.adamczyk@polsl.pl

[2] Department of Environmental Monitoring, Central Mining Institute, 40-166 Katowice, Poland; bbialecka@gig.eu

\* Correspondence: mcempa@gig.eu

**Abstract:** The aim of this study was to examine the physico-chemical and phase characteristics of ash obtained in the process of the combustion of Polish poultry manure in a laboratory reactor with a bubbling fluidized bed. Three experiments, differing in the grain size and morphology of the raw material, the method of its dosing and the type of fluidized bed, were carried out. The contents of the main chemical components and trace elements in the obtained ash samples were determined using WDXRF, and the phase composition was examined through the XRD method. The morphology and the chemical composition of grains in a given micro-area using the SEM/EDS method were also investigated. The highest concentration of phosphorus (from 28.07% wt. to 29.71% wt. as $P_2O_5$ equivalent), the highest proportion of amorphous substance (from 56.7% wt. to 59.0% wt.) and the lowest content of unburned organic substance (LOI from 6.42% to 9.16%) (i.e., the best process efficiency), was obtained for the experiment in which the starting bed was quartz sand and poultry manure was fed to the reactor in the form of pellets. It has been calculated that in this case, the amorphous phase contains more than half of the phosphorus. The method of carrying out the combustion process has a significant impact on the phase composition and, consequently, on the availability of phosphorus.

**Keywords:** ash; poultry manure; phosphorus; fluidized bed reactor

## 1. Introduction

The production and consumption of poultry (e.g., mainly broiler chickens, egg-laying hens, turkey) meat and eggs are constantly growing in Europe, and Poland is one of the leading producers of these goods. In 2019, a total of 13.3 million tonnes of poultry meat was produced in the European Union, of which Poland had the largest share among European countries—accounting for 19.5% [1]. It should be emphasized that in the years 2000–2019, there was a significant increase in the production of chicken in Poland, from 48.2 million to 178.3 million [2,3]. This trend, which has been maintained in recent years, indicates that the share of chicken in the total poultry production is more than 90%. Assuming that the amount of manure from one bird is approximately 100 g/day [4], it is estimated that approximately 6.5 million tonnes of fresh chicken manure is produced per year. Such a large mass of waste could represent a threat to the environment and requires appropriate technical and legal actions. There are a few possibilities for the effective management of poultry manure, among which the following should be mentioned: its use for fertilization in agriculture, for the reclamation of soil–which is lacking in organic matter, for energy generation, or as a fodder additive for other groups of animals, e.g., beef cattle or fish (this method is not allowed in Poland).

Poultry manure, which is a mixture of poultry droppings, feathers and bedding material, has been used as fertilizer for many years due to its high nutrient content. However, the current legal acts in force [5–8] prevent the unlimited use of this type of

waste. This is mainly due to the possibility of water pollution and eutrophication [9,10]. The use of poultry manure as fertilizer can also generate greenhouse gas emissions due to the release of methane and ammonia into the atmosphere [11,12].

In recent years, much attention has been paid to the use of animal bio-waste in the process of anaerobic digestion for the production of biogas [13,14]. As indicated by studies [15–18], pre-treated manure and poultry litter can be thermally converted as their calorific values are equivalent to low-rank coals [19]. There are many methods for the thermal conversion of biomass, including pyrolysis, gasification, combustion or co-combustion with coal. The latter solution is now becoming increasingly popular due to the minimized use of fossil fuels. Additionally, adding coal to the manure allows for greater control and stabilization of the combustion process. Moreover, the existing coal-fired power plants do not require large capital investments to be successfully used during the simultaneous combustion of both substances [20]. The processes of pyrolysis and combustion of manure to obtain thermal energy are of particular interest. The possibility of reducing carbon dioxide emissions by burning poultry manure should also be taken into account. In addition, there are several studies that analyze the combustion of poultry litter in fluidized bed combustion chambers, which are considered suitable for low calorific fuels [21].

The analysis of the literature on the manure combustion process in fluidized bed boilers indicates a number of advantages of this process but also highlights the aspects that determine the effectiveness and technological efficiency of its application [10,22–26]. It has been observed that due to the high transport costs caused by the high moisture content, the combustion of the manure is economically justified only in the vicinity of poultry farms. In addition, the ash produced in the combustion process can be a valuable material when used as fertilizer, with slow-release phosphorus (P) and potassium (K) [27]. It has been found that the composition of manure and its humidity significantly influence the course of the combustion process [22]: (i) burning poultry manure with a moisture content higher than 25% may adversely affect the mechanism of the screw feeder and, therefore the stability of the combustion process; (ii) high chlorine content in poultry waste may lead to boiler corrosion; (iii) high concentrations of P and K in manure, which are favourable for the recycling of ash as a soil conditioner, may be detrimental to the operation of the fluidized bed combustion chamber. Researchers emphasize the relative ease of controlling the combustion process, in particular [22,23]: (i) the combustion of manure in a fluidized bed boiler is characterized by relatively easy control of $SO_2$, $NO_x$ and CO emissivity to the atmosphere. Emissions can be reduced by using secondary air and especially by increasing its turbulence; (ii) problems related to ash agglomeration leading to the loss of fluidization and to the shutdown of the plant can be reduced by adding calcium salt to the fluidized bed boiler.

The analysis of ash from the fluidized bed combustion process allowed several important conclusions to be made [28,29]: (i) phosphorus was detected only in the coarse fractions—no phosphorus was found in the fine particle fraction (<1 μm), (ii) the crystalline phosphorus-containing compound was in all cases hydroxyapatite as well as whitlockite, (iii) the amorphous phase was the source of the bioavailable form of phosphorus contained in ash.

The aim of this work was to determine the physico-chemical and phase characteristics of ashes obtained in the combustion of Polish poultry manure in a laboratory reactor with a bubbling fluidized furnace. The research also aimed to assess these ashes in terms of phosphorus bioavailability. According to the literature, the source of the bioavailable form of phosphorus is the amorphous phase. Therefore, the distribution of phosphorus and calcium, magnesium, sodium and potassium content in the crystalline and amorphous phases was analysed. The obtained results may also provide a basis for developing assumptions for the construction of an industrial plant with a fluidized bed for poultry manure combustion.

## 2. Materials and Methods

### 2.1. Poultry Manure

Samples of fresh poultry manure were collected from a Polish laying hen farm in the province of Silesia. The flock numbered 52,000 birds during the study period.

Two batches of samples were taken at different times in the life of the birds, i.e., at week 6 (sample labelled R1) and week 13 (sample labelled R2). Samples were taken at random from fresh piles of cage manure and brought to an air-dry state. Samples for testing were averaged and ground to a grain size below 0.2 mm.

### 2.2. Starter Bed

In the first stage of the research, ash from the combustion of poultry manure, which is characterised by a smaller grain diameter and density than typically used sand beds, was used as a fluidised bed. It was, therefore, possible to use lower velocities of the fluidising agent, i.e., air, which increased the residence time of material particles and gases in the fluidised bed and above the bed in the high-temperature zone.

The use of raw poultry manure for the starter of the laboratory fluidised bed reactor could have caused intense heat generation in the reactor, resulting in combustion proceeding under uncontrolled conditions in terms of maintaining the desired temperature values. For this reason, the starter of the reactor was carried out using a calcined poultry manure sample as a fluidised bed. The calcination was carried out by burning part of the raw material in a muffle furnace at 750 °C–900 °C.

In the second stage of the study, a typical mineral bed, i.e., quartz sand with a grain size ranging from 200 µm to 250 µm, was used as the fluidised bed.

### 2.3. Poultry Manure Combustion Process

The combustion of poultry manure was carried out in a laboratory reactor with a fluidized bed furnace [30]. The main part of the system was a reactor in the form of a 500 mm long quartz tube. The fluidising agent was air. The airflow rate was maintained in the range of 30 dm$^3$/min–50 dm$^3$/min. In order to heat the bed and stabilise the temperature, a resistance heating jacket connected to an autotransformer was used.

During the starter phase, a previously prepared starter bed was introduced into the reactor and heating to the set temperature was initiated. During this stage, fine dusty material particles were pneumatically transported to the dedusting equipment (cyclone and settling chamber). After the temperature stabilisation, the dosing of raw material (poultry manure) began. The combustion process was carried out at the temperature of the bed 750 ± 20 °C, which was maintained by varying the material dosage rate (within the range of 5 g/min–25 g/min).

After the combustion process, samples of ash deposited in the cyclone, settling chamber, reactor walls and fluidised bed material were collected. On this basis, a mass balance of the process was prepared.

Three experiments were conducted, which differed in the grain size and morphology of the raw material grains, the method of its dosing and the type of fluidised bed (Table 1). The poultry manure samples were used to prepare materials for combustion in bulk and pellet form. The freshly collected samples R1 and R2 were brought to an air-dry state and then ground to a grain size below 2 mm, thus becoming a loose form fuel (Figure S1). Additionally, sample R2, prepared according to the above procedure, was mixed with starch and water in a mass ratio of 100:4.3:36 and pellets of 6 mm diameter and 10 mm –30 mm length were prepared from it (Figure S2).

**Table 1.** Experimental conditions for the combustion of poultry manure in the laboratory fluidised bed reactor and list of the symbols used.

| Experiment | Material | Bed | Temperature (°C) | Product Symbol |
|---|---|---|---|---|
| E1 | R1/L | A | 750 ± 20 | E1/KO E1/C |
| E2 | R2/L | Q | 750 ± 20 | E2/KO E2/C |
| E3 | R2/P | | 750 ± 20 | E3/KO E3/C |

R1, R2—poultry manure samples taken at 6 and 13 weeks of age of the birds, respectively; L—loose material, P—pellet material, A—ash obtained from combustion of poultry manure, Q—quartz sand, C—product sample from cyclone, KO—product sample from settling chamber.

### 2.4. Analysis of Raw and Burnt Poultry Manure

For the raw poultry manure samples collected, the following points were determined: total moisture content according to PN-EN ISO 18134-2:2017-03, moisture content in the analytical sample according to PN-EN ISO 18134-3:2015-11, ash content according to PN-EN ISO 18122:2016-01, combustion heat and calorific value according to PN-EN ISO 18125: 2017-07, carbon, hydrogen and total nitrogen content according to PN-EN ISO 16948:2015-07, total sulphur content according to PN-G-04584:2001, content of main chemical components and trace elements by the wave dispersive X-ray fluorescence (WDXRF) method using PRIMUS II X-ray spectrometer and SQD semi-quantitative analysis software.

The oxygen content of the sample in the analytical state ($O^a$) was determined by calculations according to the standard PN-ISO-1928:2002.

The grain morphology was also observed using binocular and scanning electron microscopy (Hitachi SEM SU3500).

For the sampled combustion products, the following points were determined: (i) loss on ignition (LOI) by the weight method, burning the sample at 900 °C; (ii) content of main chemical components and trace elements (As, Cr, Cu, Mo, Ni, Pb, Rb, Sn, V, Y, Zn, Zr) by WDXRF method (as for raw poultry manure samples); (iii) mineral composition by X-ray diffraction (XRD) using a PANalytical AERIS 1 diffractometer with a CuKα lamp under the following conditions: voltage—40 kV, intensity—8 mA, 2theta angle step size—0.002°, time—4.84 s, 2theta angle range 4–77°; the quantitative phase composition was determined by the Rietveld method; (iv) morphology and the chemical composition of grains in a given microarea using a Hitachi SEM SU3500 scanning electron microscope working alongside an X-ray energy dispersion spectrometer UltraDry EDS Detector from ThermoFisher Scientific under the following conditions: acceleration voltage—15 keV, detector—BSE, scanning time—60 s, magnification ×500—×10,000; the photos were taken after the sample was sputtered with gold; (v) bioavailable phosphorus by the Egner-Riehm method and using citric acid as extractant i.e., the ash sample was mixed with 2% citric acid and 0.04 M calcium lactate solution at a ratio of 1 g ash: 100 mL solution, the samples were centrifuged on a rotary mixer at a speed of 80 rpm for a period of 90 min (Egner-Riehm method) or 30 min (citric acid), the samples were then centrifuged for 10 min (Eppendorf 5810) and filtered using Whatman filters with a pore size of 45 μm to obtain a clear solution; total phosphorus content was determined by the Phosphate—Phosphomolybdenum blue method using a Nanocolor 500D photometer (Macherey-Nagel GmbH & Co. KG, Düren, Germany).

### 3. Results

*3.1. Characteristics of Poultry Manure*

The poultry manure varied greatly in terms of grain size and grain morphology. Irregularly shaped crumbs predominated in the samples examined, but there were also long or elongated feathers, even of a size exceeding 1 mm. The crumbs most often had a spongy structure and were, therefore, fragile and easily broken up. A clear difference was observed between the raw materials labelled R1 and R2. Sample R2 contained a significantly

higher proportion of the fine-grained fraction as well as fibres and fragments of plumage than sample R1. Organic fibres occur separately, form aggregates or are combined with mineral crumbs (Figure S1).

Preliminary combustion tests of sample R1 were performed in a laboratory muffle furnace at the temperature range from 815 °C to 1000 °C. The results of these tests showed a linear relationship between ash content and process temperature ($R^2 = 0.99$) (Figure 1). Due to the insignificant changes in the amount of ash, it was concluded that by 815 °C, the material being tested was almost completely incinerated.

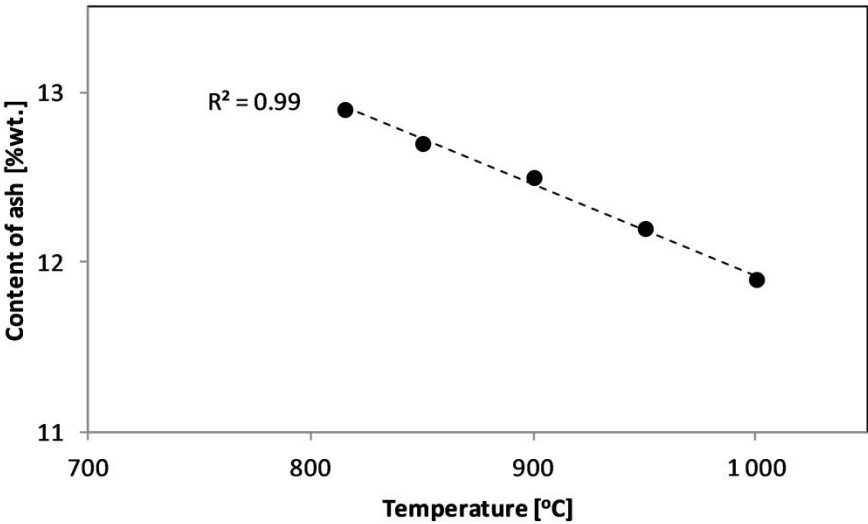

**Figure 1.** Variation in ash content of poultry manure (sample R1) according to combustion temperature (muffle furnace).

Test results for poultry manure indicated low ash content (12.8% wt.) and relatively high combustion heat and calorific value (>14,000 kJ/kg) (Table 2). A small mass of ash after the combustion process indicates a large proportion of organic matter, and thus flammable parts, which is why a high calorific value of the material studied was expected. Carbon and oxygen dominate among the main components, with N, H, P, Cl, K and Ca present in smaller amounts. Among the trace elements, the high contents of Mn and Zn (about 500 ppm) are noteworthy in comparison with the others (Table 3).

**Table 2.** Results of technical and elemental analyses and the combustion heat and calorific value of the poultry manure samples taken.

| Parameter | State | | |
|---|---|---|---|
| | **Wet** | **Analytical** | **Dry** |
| $W$ (% wt.) | 73.97 | 2.40 | - |
| $A$ (% wt.) | - | 12.80 | 13.12 |
| $Q_s$ (kJ/kg) | - | 15,504 | 15,885 |
| $Q_i$ (kJ/kg) | - | 14,334 | 14,686 |
| $S_t$ (% wt.) | - | 0.52 | 0.53 |
| $C_t$ (% wt.) | - | 37.35 | 38.27 |
| $H_t$ (% wt.) | - | 5.09 | 5.22 |
| $N_t$ (% wt.) | - | 7.55 | 7.74 |
| $O_t$ (% wt.) | - | 34.91 | 35.77 |

$W$—moisture content, $A$—ash content, $Q_s$—combustion heat in the analytical state, $Q_i$—calorific value, $C_t$—total carbon, $S_t$—total sulfur, $H_t$—total hydrogen, $N_t$—total nitrogen and $O_t$—total oxygen.

**Table 3.** Chemical composition of the poultry manure samples taken (on a dry basis).

| Main Elements | Content (% wt.) | Trace Elements | Content (ppm) |
|---|---|---|---|
| C | 38.27 | Ti | 95 |
| N | 7.74 | V | no |
| O | 35.80 | Cr | no |
| H | 5.22 | Mn | 506 |
| S | 0.53 | Co | no |
| F | 0.00 | Ni | 19 |
| Na | 0.70 | Cu | 74 |
| Mg | 0.98 | Zn | 497 |
| Al | 0.04 | Br | 7 |
| Si | 0.23 | Rb | no |
| P | 2.38 | Sr | 55 |
| S | 0.78 | Zr | 5 |
| Cl | 1.12 | Nb | no |
| K | 3.08 | Cs | 125 |
| Ca | 2.84 | Ba | 218 |
| Fe | 0.09 | Pb | no |

no—below detection limit.

### 3.2. Characteristics of Ashes from the Combustion of Poultry Manure in Laboratory Fluidized Bed Furnace

In order to develop assumptions for a larger scale combustion process, a detailed analysis of the process carried out on a laboratory scale is important. The most important parameter that was controlled during the process was the bed temperature. Detailed measurements were also taken of the masses of all solid products after combustion of the manure in the reactor, taken from the settling chamber (KO), the cyclone (C) and the reactor walls, including the masses of the starter bed and the raw material (poultry manure). In this way, it was possible to balance the masses (Table 4).

**Table 4.** Mass balance of poultry manure combustion in the fluidised bed reactor.

| Experiment | E1 | E2 | E3 |
|---|---|---|---|
| Mass balance (g) | | | |
| Starting bed * | 215.0 | 300.0 | 400.0 |
| Raw material mass | 1500.0 | 700.0 | 281.4 |
| Product mass | 435.8 | 425.7 | 420.7 |
| Mass fraction of individual ingredients in the product (% wt.) | | | |
| Bed after the combustion process | 39.0 | 79.1 | 96.2 |
| Product taken from the settling chamber (KO) | 41.6 | 15.7 | 1.4 |
| Product taken from the cyclone I | 17.6 | 2.0 | 0.1 |
| Product taken from the reactor walls (SR) | 1.9 | 3.2 | 2.3 |
| Sum KO + C | 59.2 | 17.7 | 1.5 |

* the starter bed is ash from the combustion of poultry manure (E1) or quartz sand (E2 and E3).

The results of these measurements indicated that the amount of settling chamber ash is several times greater than the amount of cyclone ash in each experiment. It is noted that the largest share of these products occurred in experiment E1, as the sum of the product from the settling chamber and the cyclone was, in this case, more than 59% wt., while the lowest share was in experiment E3, in which these products accounted for only 1.5% wt.

### 3.2.1. Chemical Composition of Ash and Grain Morphology

The combustion resulted in six ash samples, in which the dominant chemical components, occurring in amounts of several percent each, are CaO, $P_2O_5$, $K_2O$ and loss on ignition (LOI) amounts to 36.04%–62.35% wt. An exception in this respect are samples from experiment E3, in which CaO, $P_2O_5$, $K_2O$ contents are much higher (in the case of

$P_2O_5$ almost twice), and LOI contents several times lower (6.42%–9.16% wt.) than in other samples (Table 5). Components such as $SiO_2$, MgO, $Na_2O$, $SO_3$, Cl and for most samples, $Fe_2O_3$ are present in the amount of a few percent. Other chemical components do not exceed 1% wt. ($TiO_2$, $Al_2O_3$, MnO, SrO and BaO).

**Table 5.** Chemical composition (% wt.) of ashes obtained from combustion of poultry manure in three experiments E1, E3 and E3.

| Components | E1/KO | E1/C | E2/KO | E2/C | E3/KO | E3/C |
|---|---|---|---|---|---|---|
| $SiO_2$ | 1.14 | 0.88 | 0.86 | 1.08 | 3.15 | 4.11 |
| $TiO_2$ | 0.06 | 0.05 | 0.06 | 0.03 | 0.08 | 0.10 |
| $Al_2O_3$ | no | no | no | no | 1.12 | 1.42 |
| $Fe_2O_3$ | 0.82 | 0.80 | 1.18 | 1.70 | 2.51 | 4.97 |
| MnO | 0.40 | 0.36 | 0.21 | 0.27 | 0.47 | 0.42 |
| MgO | 4.78 | 3.60 | 2.36 | 4.80 | 7.75 | 6.92 |
| CaO | 19.94 | 16.63 | 11.24 | 16.20 | 26.01 | 23.08 |
| $Na_2O$ | 2.49 | 2.14 | 1.36 | 2.81 | 4.22 | 3.43 |
| $K_2O$ | 13.11 | 12.07 | 6.33 | 9.99 | 13.97 | 12.87 |
| $P_2O_5$ | 15.41 | 12.24 | 9.65 | 16.89 | 29.71 | 28.07 |
| $SO_3$ | 3.12 | 3.07 | 1.43 | 1.96 | 2.60 | 2.60 |
| Cl | 2.23 | 2.31 | 2.46 | 4.76 | 1.45 | 2.53 |
| SrO | 0.03 | 0.02 | 0.01 | 0.03 | 0.05 | 0.04 |
| BaO | no | no | no | 0.01 | 0.01 | no |
| LOI | 36.04 | 45.33 | 62.35 | 38.95 | 6.42 | 9.16 |
| Total | 99.57 | 99.50 | 99.50 | 99.48 | 99.52 | 99.72 |

no—below detection limit, KO—settling chamber, C—cyclone.

However, the content of chemical components varies between samples. It is noted that the composition of samples from pellet combustion differs in this respect from the composition of ash from the combustion of loose manure.

It is also noticeable that there is little variation in the chemical composition of ashes from the combustion of manure in loose form on different beds (E1 and E2). This variation is evidenced by slightly higher contents of: (i) MnO, $K_2O$ and $SO_3$ in the E1 experiment conducted on a bed of ash from the combustion of poultry manure, (ii) $Fe_2O_3$ and Cl in the E2 experiment conducted on a bed of quartz sand. The reasons for the large variation of LOI in these samples is difficult to identify conclusively.

On the basis of the research carried out, it was found that the contents of some elements show a high, positive, significant correlation among themselves. In particular, this concerns the main components CaO, $P_2O_5$ and $K_2O$ with other chemical components. This is clearly indicated by the high values of the coefficient $R^2$, which are, for most of the trend lines shown in Figures 2–4, above 0.90 for $p < 0.05$. These correlations apply: (i) CaO with MnO, MgO, $Na_2O$, $K_2O$, SrO; (ii) $P_2O_5$ with $SiO_2$, MgO, CaO, $Na_2O$, SrO; (iii) $K_2O$ with MnO, CaO, $SO_3$. A high, negative, significant correlation with CaO and $P_2O_5$ is only shown by LOI.

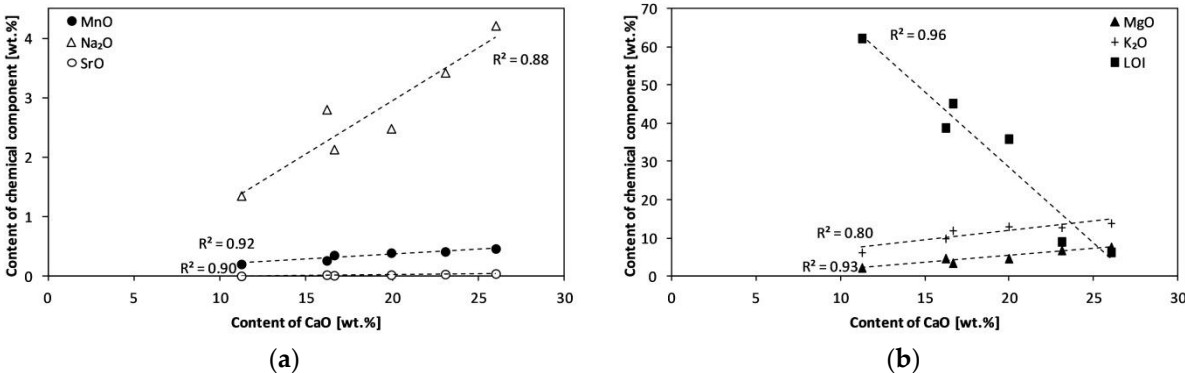

**Figure 2.** Dependence of the content of selected chemical components on the CaO content in the studied ashes from poultry manure combustion. (**a**) Dependence of the content of MnO, Na$_2$O and SrO on the CaO content. (**b**) Dependence of the content of MgO, Ka$_2$O and LOI on the CaO content.

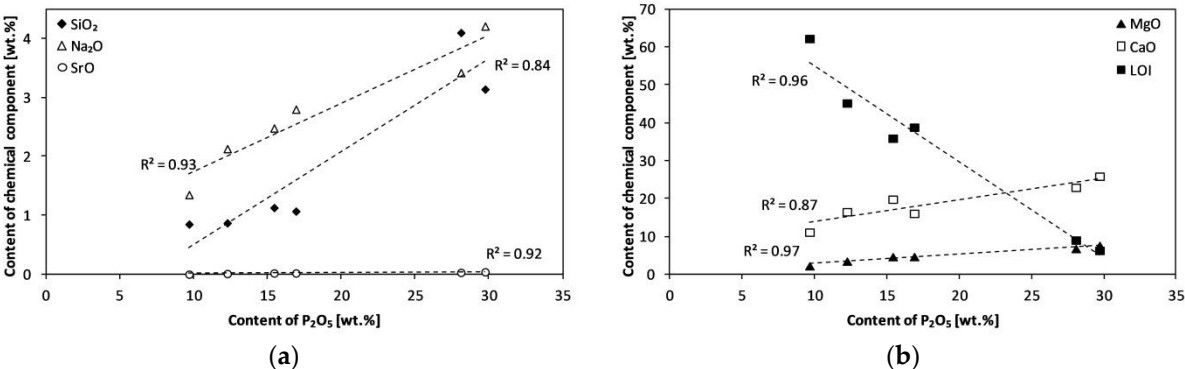

**Figure 3.** Dependence of the content of selected chemical components on the P$_2$O$_5$ content in the studied ashes from poultry manure combustion. (**a**) Dependence of the content of SiO$_2$, Na$_2$O and SrO on the P$_2$O$_5$ content. (**b**) Dependence of the content of MgO, CaO and LOI on the P$_2$O$_5$ content.

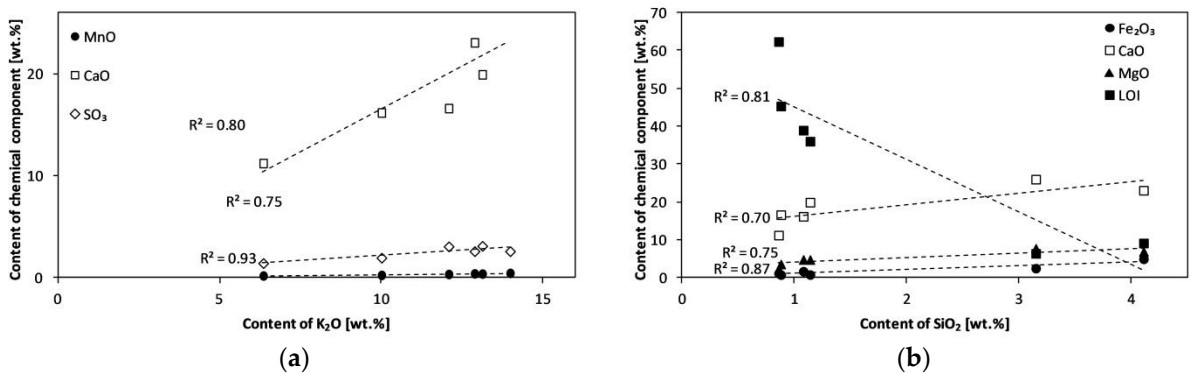

**Figure 4.** Dependence of the content of selected chemical components on the K$_2$O or the SiO$_2$ content in the studied ashes from poultry manure combustion. (**a**) Dependence of the content of MnO, CaO and SO$_3$ on the K$_2$O content (**b**) Dependence of the content of Fe$_2$O$_3$, CaO, MgO and LOI on the SiO$_2$ content.

Due to the use of quartz sand beds, the correlation coefficient between the content of individual chemical components and the content of SiO$_2$ was calculated. In this case, it was also found that the contents of chemical components such as Fe$_2$O$_3$, CaO and MgO also show a high, positive, significant correlation with SiO$_2$, while LOI correlations are high, negative and significant.

Despite detailed correlation analyses and apparent differences in chemical composition between samples from the same experiment but taken from different locations in the reactor (cyclone, settling chamber), no regularities justifying these differences were observed.

In view of, (i) the sum of the four basic chemical components ($CaO$, $P_2O_5$, $K_2O$ and LOI), fluctuating in the range 72%–89% wt.; (ii) geochemical affinity $CaO$ with $MgO$ and $K_2O$ with $Na_2O$; (iii) the mutual correlations between these components, the results for the tested samples are presented in a triangular diagram in the system $CaO + MgO$—$K_2O + Na_2O$—$P_2O_5$, considering LOI as the fourth component in the form of the proportional size of symbols (bubbles) of the individual projection points of the samples (Figure 5). The diagram shows line a, which clearly separates samples coming from the cyclone from those coming from the settling chamber.

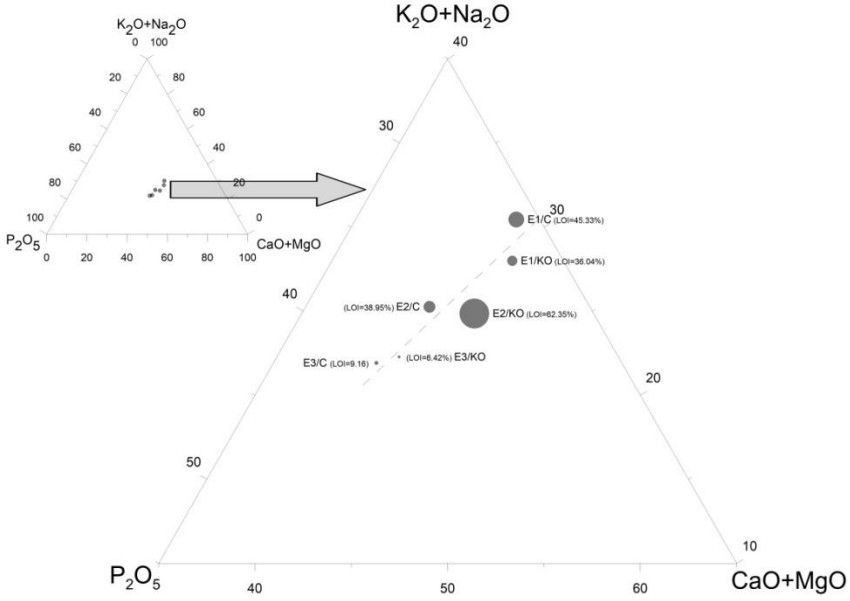

**Figure 5.** Ternary Bubble Plot in a system $CaO + MgO$—$K_2O + Na_2O$—$P_2O_5$, taking into account the LOI as a proportion of the bubble diameter.

Among the trace elements determined, Zn and Cu are clearly dominant, with contents ranging from 3280 ppm to 3790 ppm and from 583 ppm to 901 ppm, respectively (Table 6). The contents of other elements rarely exceed 100 ppm, i.e., Ni (in sample E2/KO, E2/C), Pb (in sample E2/C), Rb (in samples E1/KO, E1/C, E2/KO, E2/C) and Zr (in samples E3/KO, E3/C). Concentrations of other elements (As, Cr, Mo, Sn, V and Y) usually did not exceed 50 ppm and, in some cases, were below the detection limit.

Some variation in the content of certain elements is apparent, (i) in experiment E1, lower contents of Ni, Pb, Sn and Zr are observed than in E2 and E3; (ii) in experiment E2, higher contents are present for As, Ni and Rb compared to experiments E1 and E3; (iii) in experiment E3 significantly higher contents of Cr and Zr and significantly lower Cu are observed compared to experiments E1 and E2.

Variations in the content of the main chemical components and trace elements in the tested poultry manure ashes in individual experiments are most likely due to the conditions of their conduct—two types of starting deposits and two forms of preparation of the samples fed to the reactor (loose and pellets), which is related to, (i) residence time of combusted particles in the combustion chamber; (ii) the influence of the chemical components of the bed on the chemical composition of ashes from the combustion of manure.

It was therefore observed that the highest phosphorus concentration and the lowest unburned organic matter content (lowest LOI), i.e., the best efficiency, were obtained for

experiment E3, in which the starting bed was quartz sand and the poultry manure was fed to the reactor in the form of pellets.

**Table 6.** Trace element concentrations (ppm) in the investigated ashes from poultry manure combustion.

| Element | E1/KO | E1/C | E2/KO | E2/C | E3/KO | E3/C |
|---------|-------|------|-------|------|-------|------|
| As | 3 | 5 | 21 | 14 | 7 | no |
| Cr | no | no | 39 | 12 | 96 | 84 |
| Cu | 818 | 901 | 802 | 895 | 638 | 583 |
| Mo | 23 | 18 | no | 14 | 15 | 10 |
| Ni | 34 | 41 | 184 | 210 | 70 | 75 |
| Pb | 5 | 6 | 95 | 156 | 52 | 78 |
| Rb | 113 | 121 | 141 | 205 | 94 | 88 |
| Sn | no | 27 | no | 33 | 40 | 25 |
| V | 24 | 31 | 30 | 18 | 14 | 18 |
| Y | 10 | 8 | 2 | 1 | 6 | 8 |
| Zn | 3280 | 3790 | 3610 | 3540 | 3600 | 3420 |
| Zr | 30 | 23 | 51 | 67 | 209 | 390 |

no—below detection limit.

The morphology (Figure S3) and chemical composition of 20 randomly selected grains in the sample of ash obtained in experiment E3, both taken from the cyclone (Figure 6a) and the settling chamber (Figure 6b) by EDS, were also examined. In the case of the main elements, i.e., calcium, phosphorus and potassium, the difference between the content of a given element determined by the XRF method and the average content calculated on the basis of EDS tests does not exceed 15%. The chemical composition of the ash grains was compared with the theoretical composition of the main phosphate minerals identified in the tested ash samples (Figure 7). It can be observed that the grains selected for the analysis consisted mainly of amorphous substances.

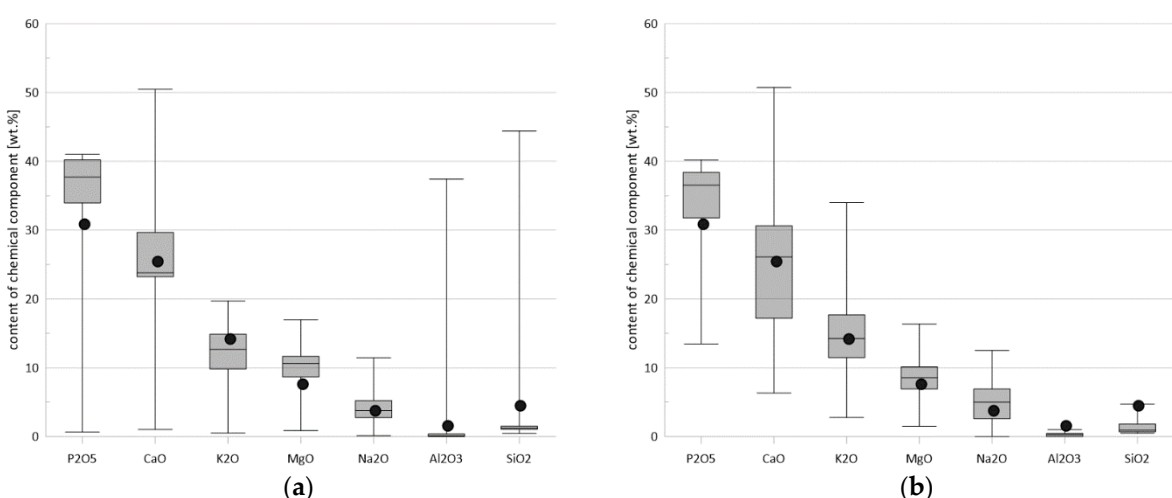

**Figure 6.** Chemical composition of selected ash grains in the samples of ash obtained in experiment E3 determined by the EDS method. (**a**) Chemical composition of selected ash grains in sample E3/C. (**b**) Chemical composition of selected ash grains in sample E3/KO. (centre line—median, box edges—percentiles 25%/75%, whiskers—minimum/maximum) and XRF method (spot) (note: element C was excluded from the analysis).

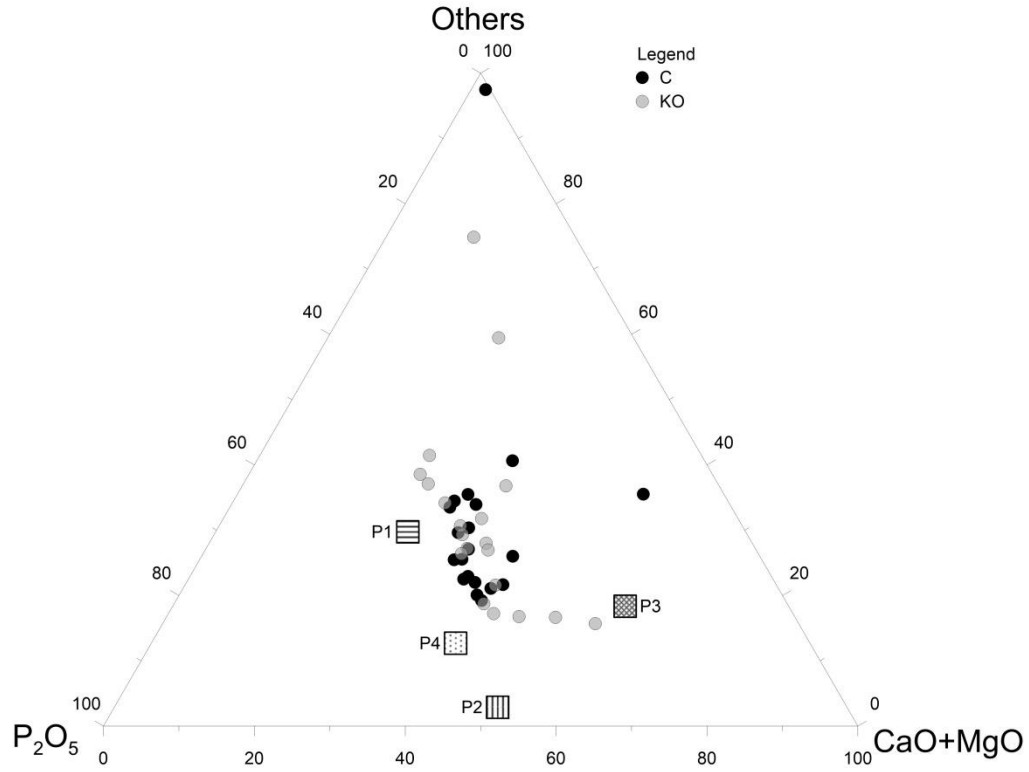

**Figure 7.** Ternary Bubble Plot in the system MgO + CaO—Others—$P_2O_5$. Annotations: C—ash grain composition of sample E3/C, KO—ash grain composition of sample E3/KO, Pi—theoretical chemical composition of selected phosphorous phases, P1-potassium magnesium phosphate (V), P2-nonacalcium magnesium sodium heptakis (phosphate (V)), P3-nagelschmidtite, P4-wopmayite.

### 3.2.2. Phase Composition of Ash

X-ray diffraction (XRD) identified the following phase components in all ash samples (Table 7, Figure S4): potassium magnesium phosphate (V), nonacalcium magnesium sodium heptakis (phosphate (V)), nagelschmidtite, periclase and sylvine. Apatite (except for sample E3/C), wopmayite, arcanite (except for sample E2/KO) and calcite (except for sample E2/C) were also present in most of the samples examined. Only some samples contained whitlockite (E1/KO, E1/C and E2/C), calcium iron magnesium hydrogen phosphate (E1/C and E2/C) and metathenardite (only in samples from experiment E1).

Diffractograms of all samples show a high background in the range of 2 theta 15–30°, which clearly indicates the presence of an amorphous substance in the ashes studied, associated with the presence of unburned organic matter, as well as the glass.

Among the identified phases, due to their chemical composition, two types of components can be distinguished, i.e., (i) those containing phosphorus, i.e., potassium magnesium phosphate (V), nonacalcium magnesium sodium heptakis (phosphate (V)), nagelschmidtite, wopmayite, apatite, whitlockite, calcium iron magnesium hydrogen phosphate, (ii) other components, i.e., arcanite, calcite, m etathenardite, periclase, sylvine.

Thus, phosphorus in crystalline form occurs in combination with the following metals, (i) calcium (apatyt, nagelschmidtite), (ii) calcium and magnesium (whithlockite) and sodium (nonacalcium magnesium sodium heptakis (phosphate (V)), (iii) potassium and magnesium (potassium magnesium phosphate (V)), (iv) magnesium and iron (calcium iron magnesium hydrogen phosphate) and sodium and manganese (wopmayite).

It was found that among the crystalline phases, the phases containing phosphorus—magnesium phosphate (V), nonacalcium magnesium sodium heptakis (phosphate (V)) and nagelschmidtite, whose amounts are usually several percent each, and in some samples even above 20% wt. (samples from experiment E2). Their total amount was in the range of 25.8% wt. to 59.9% wt. The sum of all crystalline phases containing phosphorus ranged

from 37.7% wt. to 70.2% wt. It is noteworthy that the highest amounts of these phases were found in samples from experiment E2, while in experiments E1 and E3, these amounts were comparable (37.7%–42.3% wt.).

**Table 7.** Phase composition (wt. %) of ashes from poultry manure using the XRD method.

| Name | E1/KO | E1/C | E2/KO | E2/C | E3/KO | E3/C |
|---|---|---|---|---|---|---|
| **Phosphorous Phases** | | | | | | |
| Potassium Magnesium Phosphate (V) | 13.6 | 11.6 | 25.5 | 21.0 | 11.7 | 11.3 |
| Nonacalcium Magnesium Sodium Heptakis (phosphate (V)) | 8.6 | 1.2 | 12.7 | 24.6 | 15.3 | 15.5 |
| Nagelschmidtite | 12.3 | 13.0 | 18.0 | 14.3 | 9.8 | 11.5 |
| Total—Main Phosphorous Phase | 34.5 | 25.8 | 56.2 | 59.9 | 36.8 | 38.3 |
| Wopmayite | 0.8 | 4.2 | no | 8.3 | 3.0 | 4.0 |
| Apatite | 3.1 | 2.8 | 0.4 | 0.6 | 0.1 | no |
| Whitlockite | 1.8 | 4.1 | no | 1.2 | no | no |
| Calcium Iron Magnesium Hydrogen Phosphate | no | 0.8 | no | 0.2 | no | no |
| Total—All Phosphorous Phases | 40.2 | 37.7 | 56.6 | 70.2 | 39.9 | 42.3 |
| **Other Phases** | | | | | | |
| Arcanite | 6.2 | 6 | no | 0.6 | 0.1 | 0.3 |
| Calcite | 0.5 | 0.7 | 0.2 | no | 0.2 | 0.1 |
| Metathenardite | 5.4 | 8.2 | no | no | no | no |
| Periclase | 0.8 | 1.2 | 0.1 | 0.5 | 0.3 | 0.1 |
| Sylvine | 4.1 | 3.8 | 2.0 | 7.4 | 0.5 | 0.5 |
| Total—other phases | 17.0 | 19.9 | 2.3 | 8.5 | 1.1 | 1.0 |
| Total—Crystalline Phase | 57.2 | 57.6 | 58.9 | 78.7 | 41.0 | 43.3 |
| Total—Amorphous Phase | 42.8 | 42.4 | 41.1 | 21.3 | 59.0 | 56.7 |
| Total | 100.0 | 100.0 | 100.0 | 100.0 | 100.0 | 100.0 |

The proportion of other crystalline components varied considerably. The highest amounts were shown for experiment E1 (about 18.5% wt.), while the lowest was in experiment E3 (about 1% wt.).

The proportion of amorphous substances ranged from 21.3% wt. to 59.0% wt., with significantly higher contents in samples from experiment E3 (56.7%–59.0% wt.) than in E1 and E2 (21.3%–42.8% wt.).

Practically no relationships were shown between phase components among, as well as between, phase components and chemical components. The calculated values of correlation coefficients were mostly insignificant ($p > 0.05$), even at high values. This is most likely due to the sample population being too small.

On the basis of the quantitative XRD analysis of all crystalline phases, taking into account their stoichiometric formulations, the shares of the main chemical components in the ashes from experiment E3, which had the best process efficiency, were calculated. The results of these calculations were used to estimate the proportion of the main chemical components in the amorphous phase, taking into account the results of the chemical composition determined by the XRF method (Table 8).

A higher proportion of phosphorus, calcium, potassium and sodium was observed in the amorphous phase than in the crystalline phase.

**Table 8.** Distribution of the main elements between the crystalline and amorphous phases (% wt.) of ashes obtained from the combustion of poultry manure (experiment E3).

| Component | Content in the Crystalline Phase * | | Content in the Amorphous Phase * | | Total Content ** | |
|---|---|---|---|---|---|---|
| | E3/KO | E3/C | E3/KO | E3/C | E3/KO | E3/C |
| $P_2O_5$ | 13.8 | 14.0 | 15.9 | 14.1 | 29.7 | 28.1 |
| CaO | 8.6 | 8.9 | 17.4 | 14.2 | 26.0 | 23.1 |
| $K_2O$ | 3.8 | 3.8 | 10.2 | 9.1 | 14.0 | 12.9 |
| MgO | 3.9 | 3.6 | 3.9 | 3.3 | 7.8 | 6.9 |
| $Na_2O$ | 0.6 | 0.6 | 3.6 | 2.8 | 4.2 | 3.4 |
| Sum | 30.7 | 30.9 | 51.0 | 43.4 | 81.7 | 74.4 |

* calculated value, ** measured value by XRF.

### 3.2.3. Bioavailable Forms of Phosphorus

The content of bioavailable forms of phosphorus was determined in the ash from experiment E3 by the Egner-Riehm method, as well as the extraction in 2% citric acid (Table 9).

**Table 9.** Content of bioavailable forms of phosphorus (% wt.) in ash obtained from the combustion of poultry manure in experiment E3.

| Sample | Egner-Riehm Method | Citric Acid Method |
|---|---|---|
| E3/KO | 0.65 | 21.31 |
| E3/C | 0.80 | 20.50 |

## 4. Discussion

The chemical composition and dry matter content in studied laying hen manure were typical for this type of material and similar to the results of other authors in Poland [31,32]. The content of phosphorous in tested samples was 2.38% dry wt. Parker and Perkins [33] tested hen manure samples from seven Northeast Georgia counties. The average content of phosphorous amounted to $1.91 \pm 0.48\%$ dry wt. Samples taken from a large German farm were also characterized by a high content of phosphorus [34]. Of all the poultry species, feed for the laying hens typically contains much more P relative to the requirement because of concerns of inadequate mineralization of eggshells and skeletal abnormalities [27]. Deniz et al. [35] evaluated the effects of diets of white-laying hens on performance, egg quality, feed cost and selected manure parameters. The content of phosphorous in manure fluctuated between 1.03% and 1.24% dry wt. The amount of total P in poultry manure varies with the diet. Phosphorus in manure can be reduced by feeding the birds less P or treating feed to improve phosphorus utilization efficiency [27].

The breeding method can also affect grain size and morphology, resulting in the presence of both irregular crumbs and elongated particles (fibrous and feathery in shape) of different proportions in the poultry manure. Crumbs of organic origin, i.e., fragments of tissues and feathers, due to their flexibility, were more mechanically resistant, so during physical operations (drying, sieving, mixing), their dimensions did not change, and during combustion, they are unlikely to fragment (gradual combustion). The combustion of organic particles may require a longer time in the high-temperature zone, and given that they are very light, this may indicate process problems.

The low LOI content in ash from experiment E3 clearly indicates that the poultry manure delivered to the fluidised bed reactor in the form of pellets was much better burnt than the bulk form of manure. It can, therefore, be concluded that the residence time of the pellets in the combustion zone was much longer than that of the manure particles in the loose form.

In spite of detailed analyses and slight differences in phase composition between samples originating from the same experiment but taken from different places in the

reactor (cyclone, settling chamber), as in the consideration of chemical composition, no regularities justifying these differences were observed.

The phase composition of the tested samples is shown in the triangular diagram in the system Amorphous substance—Phosphorous phases—Other phases, considering $P_2O_5$ as the fourth component in the form of proportional size symbols (bubbles) of individual projection points of the samples (Figure 8).

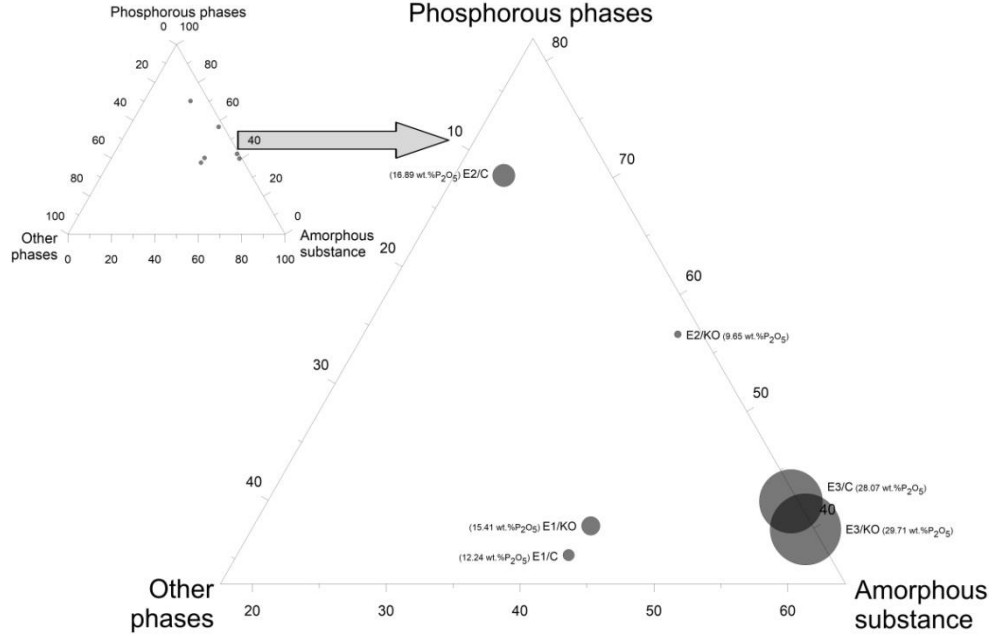

**Figure 8.** Ternary Bubble Plot in Amorphous substance—Phosphorous phases—Other phases system, considering $P_2O_5$ as a proportion of the bubble diameter.

From this diagram, it can be clearly concluded that among the samples tested, the ashes from experiment E3 will show the best availability of phosphorus, as their projection points are closest to the apex amorphous substance. At the same time, according to their chemical composition, these ashes showed the highest phosphorus content and the lowest LOI content.

The chemical compositions of ashes obtained from the combustion of poultry manure from farms in Poland in a fluidised bed reactor (experiment E3) and in a fixed bed reactor, i.e., in a laboratory muffle furnace [31] were compared. The total content of the main elements (i.e., phosphorus, calcium and potassium) in the ashes obtained from the combustion of poultry manure as pellets in a fluidised bed (experiment E3) and in a muffle furnace are similar (Figure 9).

However, the difference in phosphorus distribution between the amorphous and crystalline phases is important. It is believed that the amorphous phase is the main carrier of phosphorus which is easily soluble and therefore potentially available to plants [29,31]. It has been calculated that in ash obtained from fluidised bed combustion, there is approximately three times more phosphorus in the amorphous phase than in ash from fixed bed combustion obtained in a similar range of temperature prevailing in the bed (Figure 10).

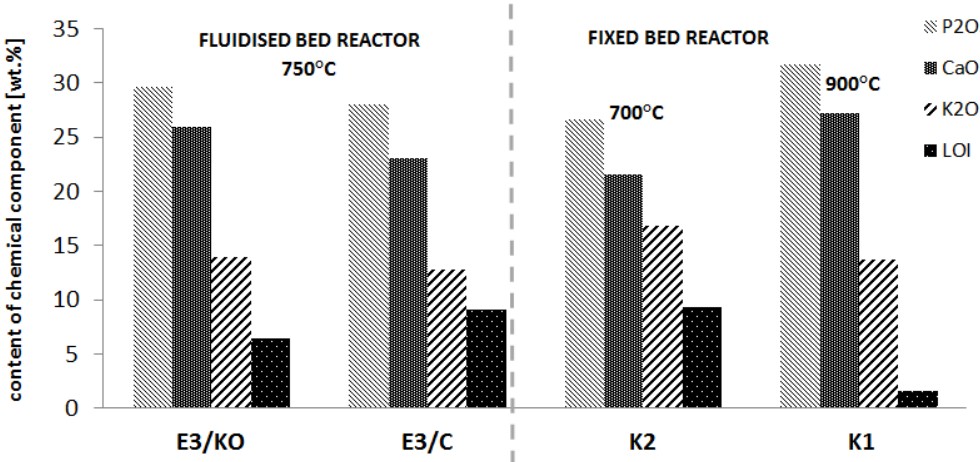

**Figure 9.** Comparison of the composition of ash obtained from the combustion of poultry manure in a fluidised bed reactor and a fixed bed reactor (E3/KO, E3/C—own work; K2, K1—[31]).

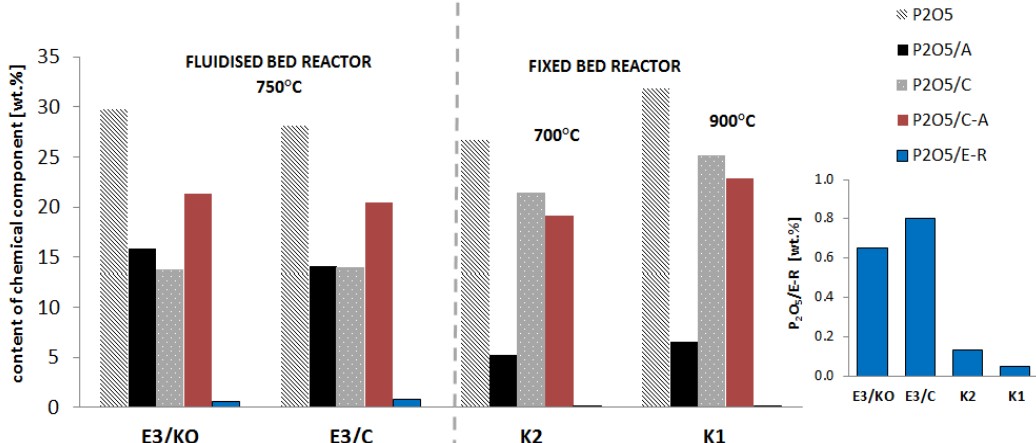

**Figure 10.** Comparison of the content of different forms of phosphorus in ash obtained from the combustion of poultry manure in a fluidised bed reactor (samples E3/KO, E3/C—own work) and a fixed bed reactor (samples K2, K1—[31]). Annotations: $P_2O_5$/A—phosphorus content in the amorphous phase, $P_2O_5$/C—phosphorus content in the crystalline phase, $P_2O_5$/C-A—the content of phosphorus easily soluble in citric acid, $P_2O_5$/E-R—the content of phosphorus easily soluble in 0.04 M calcium lactate solution.

We observed the following correlations between different forms of phosphorus in Polish ash from the combustion of poultry manure (in both a fluidised and fixed bed reactor), (i) high ($R^2 = 0.88$, $p < 0.05$), positive correlation between phosphorus content, which is easily soluble in 0.04 M calcium lactate solution ($P_2O_5$/E-R) and the phosphorus content in the amorphous phase ($P_2O_5$/A); (ii) high ($R^2 = 0.93$, $p < 0.05$) negative correlation between $P_2O_5$/E-R and phosphorus content in the crystalline phase ($P_2O_5$/C) (Figure 11); (iii) high ($R^2 = 0.99$, $p < 0.05$), positive correlation between the content of phosphorus which is easily soluble in citric acid ($P_2O_5$/C-A) and the total phosphorus content of the ash ($P_2O_5$), as in the research of Staroń et al. [36] (Figure 12); (iv) no significant statistical relationship between $P_2O_5$/C-A and $P_2O_5$/A or $P_2O_5$/C.

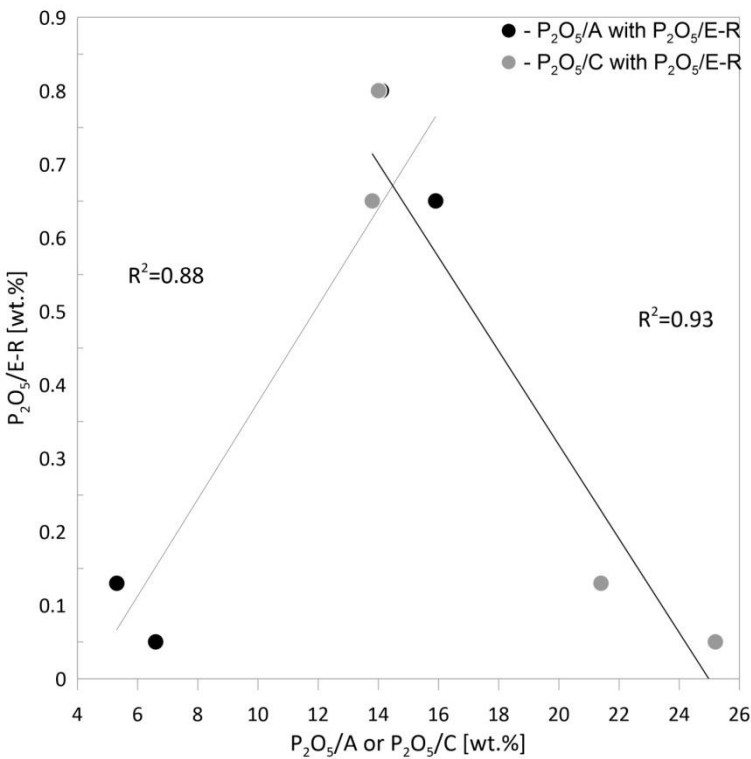

**Figure 11.** The relationship between the phosphorus content in the amorphous phase ($P_2O_5$/A) or crystalline ($P_2O_5$/C) and the content of phosphorus soluble in a 0.04 M solution of calcium lactate ($P_2O_5$/E-R).

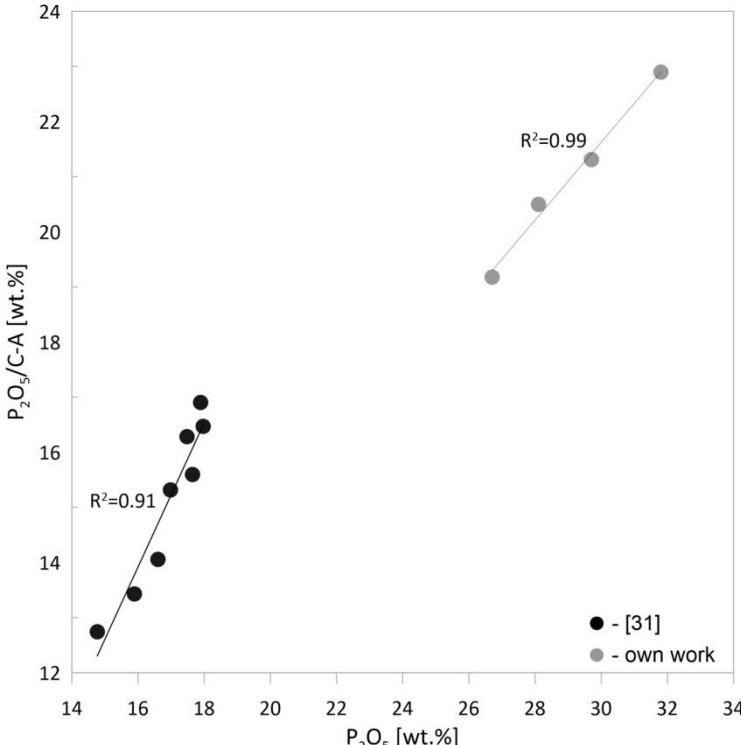

**Figure 12.** The relationship between the total phosphorus content ($P_2O_5$; determined by XRF) and the content of phosphorus soluble in 2% citric acid ($P_2O_5$/C-A).

The content of bioavailable forms of phosphorus is of particular importance in the case of using ashes as a raw material for the production of phosphate fertilisers. Various methods are used for its determination, including Egner-Riehm, Olsen, Mehlich-3 methods or extraction in 2% citric acid [37]. It is believed that after the phosphorus deficiency event, the plant secretes various substances through the root system, such as organic acids, phenolic compounds and mucilages, as well as phosphatases and other enzymes that facilitate the release of phosphorus from sparingly soluble compounds. Organic acids secreted by plants include, among others, citric and lactic acid [38,39].

However, phosphorus solubility in citric acid is not always representative of the fertilizing value. Pot and field tests give a better indication of the real bioavailability of fertilizers [40]. The Egner-Righma method is considered a suitable method for the assessment of bioavailable phosphorus content in ashes from thermal biomass conversion [41].

## 5. Conclusions

As a result of this research, the chemical and phase compositions of the ashes were determined, and the content of the bioavailable forms of phosphorus were determined. The following conclusions were made in the course of this study:

1. Poultry manure is a very heterogeneous material in terms of grain size and grain morphology, which is strongly influenced by how and when the birds breed. The grain morphology of the manure also has a strong influence on the combustion process. Poultry manure pelletizing, i.e., changing its morphology, significantly improves the efficiency of the combustion process as well as the chemical and phase properties of the ash obtained. The main product of the process is ash collected in the settling chamber.
2. The higher combustion efficiency of poultry manure in pellet form over the loose form is evidenced by the high phosphorus content (about 29% wt.) and the low content of ignition losses (LOI about 8%).
3. The phosphorus content of the crystalline phase is similar to that of the amorphous phase, with a tendency for a higher concentration in the latter.
4. The high proportion of amorphous matter in the ash obtained from the combustion of pellets (about 58% wt.) indicates a high potential for the bioavailability of phosphorus in this raw material.
5. In ash from fluidised bed combustion, there was approximately three times more phosphorus in the amorphous phase than in the ash from the fixed bed combustion obtained in a similar temperature range.

**Supplementary Materials:** The following are available online at https://www.mdpi.com/article/10.3390/min11070785/s1, Figure S1: Images of the grain morphology of poultry manure samples taken and prepared for the combustion process (after grinding): (a,b)—images from the binocular of sample R1; (c,d)—images from the binocular of sample R2; (e,f)—SEM images of sample R1; (g,h)—SEM images of sample R2, Figure S2: Images of poultry manure sample R2: (a) general view of freshly collected sample before grinding, (b,c) general view of pellets, Figure S3: SEM images of selected grains (a,b,e,g) and EDS spectra of selected micro-areas of grains (b,d,f,h) of ash obtained in E3 experiment, Figure S4. Diffractograms of the tested poultry manure ashes incinerated in a fluidised bed reactor. Annotations: P1-potassium magnesium phosphate(V), P2-nonacalcium magnesium sodium heptakis(phosphate(V), P3-nagelschmidtite, P4-wopmayite, P5-apatite, P6-whitlockite, P7-calcium iron magnesium hydrogen phosphate, A-arcanite, C-calcite, M-metathenardite, R-periclase, S-sylvine.

**Author Contributions:** Conceptualization, B.B. and M.C.; methodology, B.B., Z.A. and M.C; validation, Z.A., M.C. and B.B.; formal analysis, M.C. and Z.A.; investigation, Z.A. and M.C.; resources, M.C.; data curation, Z.A. and M.C.; writing—original draft preparation, Z.A. and M.C.; writing—review and editing, Z.A., M.C. and B.B; visualization, Z.A. and M.C; supervision, B.B.; project administration, M.C.; funding acquisition, B.B. All authors have read and agreed to the published version of the manuscript.



**Funding:** The paper has been prepared in the frames of the project: "Design of a product for substitution of phosphate rocks—DEASPHOR" co-funded by the National Centre for Research and Development (ERA-MIN2/DEASPHOR/2/2019) under the ERA-MIN2 Joint Call 2017.

**Conflicts of Interest:** The authors declare no conflict of interest.

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
