# Peer review of "Phosphorus-Rich Ash from Poultry Manure Combustion in a Fluidized Bed Reactor"

_minerals, doi:10.3390/min11070785_

Round 1

Reviewer 1 Report

The study investigated the characteristics of poultry manure ashes from a combustion process. They concluded that grain morphology was influenced by the time and type of birds breeding as well as to the combustion process. Moreover, P content was higher in the pellet form as compared to the loose form. The sections were clear and easy to follow. Please find attached the document with my suggestions and comments.  

Reviewer 2 Report

The paper investigated chemical and physical structure of combusted poultry manure. The author used various methods to compare the characteristics of manure under different combustion temperature such as SE, WDXRF and XRD. This study could provide some idea for relevant studies. However, there have some problem:

  1. Materials and methods part is too long and some of information not highlighted. Some of information could listed in a table and relevant information move to appendix.
  2. Only important table and figure listed in the main content
  3. It is meaningful to compare similar study about manure in different counties, due to different fodder and species.
